# Viral infection and transmission in a large, well-traced outbreak caused by the SARS-CoV-2 Delta variant

Baisheng Li[1,2,8], Aiping Deng[1,2,8], Kuibiao Li[3,8], Yao Hu[1,2,8], Zhencui Li[1,2,8], Yaling Shi[4,8], Qianling Xiong[1,2,5], Zhe Liu[1,2,5], Qianfang Guo[1,2], Lirong Zou[1,2], Huan Zhang[1,2], Meng Zhang[1,2], Fangzhu Ouyang[1,2], Juan Su[1,2], Wenzhe Su[3], Jing Xu[1,2], Huifang Lin[1,2,5], Jing Sun[1,2,5], Jinju Peng[1,2,5], Huiming Jiang[1,2,5], Pingping Zhou[1,2,5], Ting Hu[1,2], Min Luo[1,2], Yingtao Zhang[1,2], Huanying Zheng[1,2], Jianpeng Xiao[1,2,5], Tao Liu[1,2,5], Mingkai Tan[4], Rongfei Che[1,2], Hanri Zeng[1,2], Zhonghua Zheng[1,2], Yushi Huang[1,2], Jianxiang Yu[1,2], Lina Yi[1,2,5], Jie Wu[1,2], Jingdiao Chen[1,2], Haojie Zhong[1,2], Xiaoling Deng[1,2], Min Kang[1,2], Oliver G. Pybus[6], Matthew Hall[7], Katrina A. Lythgoe[7], Yan Li[1,2✉], Jun Yuan[3✉], Jianfeng He[1,2✉] & Jing Lu[1,2,5✉]

The SARS-CoV-2 Delta variant has spread rapidly worldwide. To provide data on its virological profile, we here report the first local transmission of Delta in mainland China. All 167 infections could be traced back to the first index case. Daily sequential PCR testing of quarantined individuals indicated that the viral loads of Delta infections, when they first become PCR-positive, were on average ~1000 times greater compared to lineage A/B infections during the first epidemic wave in China in early 2020, suggesting potentially faster viral replication and greater infectiousness of Delta during early infection. The estimated transmission bottleneck size of the Delta variant was generally narrow, with 1-3 virions in 29 donor-recipient transmission pairs. However, the transmission of minor iSNVs resulted in at least 3 of the 34 substitutions that were identified in the outbreak, highlighting the contribution of intra-host variants to population-level viral diversity during rapid spread.

[1] Guangdong Provincial Center for Disease Control and Prevention, Guangzhou, Guangdong, China. [2] Guangdong Workstation for Emerging Infectious Disease Control and Prevention, Chinese Academy of Medical Sciences, Guangzhou, Guangdong, China. [3] Guangzhou Center for Disease Control and Prevention, Guangzhou, Guangdong, China. [4] Guangzhou 8th People's Hospital, Guangzhou, Guangdong, China. [5] Guangdong Provincial Institution of Public Health, Guangzhou, Guangdong, China. [6] Department of Zoology, University of Oxford, Oxford OX1 3SZ, UK. [7] Big Data Institute, Nuffield Department of Medicine, University of Oxford, Old Road Campus, Oxford OX3 7LF, UK. [8] These authors contributed equally: Baisheng Li, Aiping Deng, Kuibiao Li, Yao Hu, Zhencui Li, Yaling Shi. ✉email: 13580581074@126.com; yuanjuncom@163.com; hjf@vip.sina.com; Jimlu0331@163.com

During the global spread of the COVID-19, genetic variants of the SARS-CoV-2 virus have emerged. Some variants have increased transmissibility or could exhibit an increased propensity for escape from host immunity, and therefore pose an increased risk to global public health[1–3]. An emerging genetic lineage, B.1.617, has gained global attention and has been dominant in the largest outbreak of COVID-19 in India since March 2021. One descendent lineage, B.1.617.2, which carries spike protein mutations L452R, T478K, and P681R, accounts for ~28% sequenced cases in India and has rapidly replaced other lineages to become dominant in multiple regions and countries (https://outbreak.info/)[4]. Lineage B.1.617.2 has been labeled a variant of concern (VOC) and given the name Delta (https://www.who.int/activities/tracking-SARS-CoV-2-variants). Data on the virological profile of the Delta VOC is needed.

On 21 May 2021 the first local infection of the Delta variant in Guangzhou, Guangdong, China was identified. As of the early epidemic in China in January 2020[5], a suite of comprehensive interventions have been implemented to limit transmission, including population screening, active contact tracing, and centralized quarantine/isolation. However, in contrast to the limited level of onward transmission observed in Guangdong in early 2020[5], successive generations of virus transmission were observed in the 2021 outbreak of the Delta variant in the region. Here, we investigated epidemiological and genetic data from the well-traced outbreak in Guangdong in order to characterize the virological and transmission profiles of the Delta variant. We discuss how intervention strategies may need to be adjusted to cope with the changes in viral shedding interval and viral load trajectories of this emerging variant.

## Results

**Viral trajectories of Delta variant infections.** A total of 167 local infections were identified during the outbreak, starting with the first index case identified on 21 May 2021 and ending with the last case reported on 18 June 2021 (Fig. 1a). All cases could be epidemiologically or genetically traced back to the first index case (Fig. 1b). One notable epidemiologic feature of the Delta variant is a shorter serial interval compared with infection with early Wuhan-like strains or other VOC variants[6–8]. However, critical parameters before and after the illness onset remain poorly known, including when the viruses can be first detected in an individual after exposure, the duration of viral shedding, and how infectious infected individuals are.

We first investigated the data from the quarantined individuals in this outbreak and compared it to data from the early 2020 epidemic caused by A/B genetic lineages (Pango nomenclature[9]) strains. The centrally-quarantined individuals were the close contacts of confirmed cases. Once a new infection was identified, their close contacts were immediately traced, centrally isolated, and underwent daily PCR testing. The dataset from quarantined individuals allowed us to determine the time interval in the infected individuals between exposure and when viral loads were first detectable by PCR. Intra-family transmission pairs were removed from our time interval analysis if the exact exposure time for the intra-family transmissions was difficult to pinpoint. Our results revealed that the time interval from exposure to the first PCR + test in the quarantined population was 6.00 days (IQR 5.00–8.00) during the 2020 epidemic ($n = 29$; peak at 5.61 days) and 4.00 days (IQR 3.00–5.00) in the 2021 Delta epidemic ($n = 46$; peak at 3.86 days; Fig. 1c).

We next evaluated viral load measurements at the time when SARS-CoV-2 was first detected by PCR in each individual. The relative viral loads of cases infected with the Delta variant ($n = 62$, Ct = 24.00 for the *ORF1ab* gene, IQR 19.00–29.00) were 1260 times higher than those for the 2020 infections with clade A/B viruses ($n = 63$, Ct = 34.31 for *ORF1ab* gene, IQR 31.00–36.00) on the day when viruses were first detected (Fig. 1d). We hypothesized a higher within-host growth rate of the Delta variant, which led to the higher observed viral loads once viral nucleotides exceeded the PCR detection threshold (Fig. 1e). Similar to results reported by Roman et al.[10], we found that samples with Ct > 30 ($<6 \times 10^5$ copies/mL viruses) did not yield an infectious isolate in-vitro. For the Delta variant infections, 80.65% of samples contained $>6 \times 10^5$ copies/mL in oropharyngeal swabs when the viruses were first detected, compared to 19.05% of samples from clade A/B infections (Supplementary Table 1). These data highlight that the Delta variant could be more infectious during the early stage of the infection (Fig. 1e).

To determine the viral RNA trajectories, longitudinal PCR testing was performed on 813 oropharyngeal swabs from 46 individuals (Supplementary Fig. 1). On average, individuals had eight positive PCR tests, before the virus was cleared (Supplementary Fig. 1). The lowest recorded Ct value for each of the 46 individuals had mean 18.51 (IQR:16.94–21.95), equivalent to ~$1.1 \times 10^9$ copies/mL. The exponential growth of viral RNA was observed after the virus became detectable (Fig. 1f). Using a generalized additive model (GAM) it was estimated the viral RNA peaked at 7.97 days and 8.86 days after exposure, for asymptomatic/mild and moderate/severe infections, respectively (Fig. 1f). The viral shedding intervals, from first PCR + test to first PCR- test, had a median duration of 17 days (IQR: 14–20). No significant difference of latent intervals or viral shedding intervals between groups stratified by symptom status was observed, possibly due to the limited sample size for each group (Table 1).

Individuals undergo a latent period after infection, during which viral titers are too low to be detected. As viral proliferation continues within host, the viral load will eventually reach detectable levels and the individual will become infectious. Knowing when an infected person can transmit is essential for designing intervention strategies that break chains of transmission. However, infectiousness is difficult to measure from clinical investigations since >50% of transmission occurs during the pre-symptomatic phase[11]. Our investigation of quarantined individuals suggests that, for the Delta variant, the average time window from exposure to the detection of the virus was ~3.86 days, and infections presented a higher transmission risk when the virus was first detected compared to earlier circulating viral lineages. The shorter incubation time (time from exposure to first PCR+) and higher viral loads during the early stages of Delta variant infections led the provincial government to adjust the previous non-pharmaceutical interventions (NPIs) to cope with these changes. The government required people leaving Guangzhou city from airports, train stations, and shuttle bus stations to show proof of a negative COVID-19 test within 72 h on June 6 and this was shortened to 48 h on June 7. In contrast, the comparable time window implemented in the 2020 epidemic was seven days. The intervention strategies, including increasing the frequency of population testing in high-risk areas, and active contact tracing to minimize viral transmission during pre-symptomatic infection, were implemented and well contained the recent Delta variant outbreaks in mainland China.

**Association between minor iSNVs transmissions and viral population diversity.** The non-pharmaceutical interventions in Guangdong mainly focus on epidemiological investigation, contact tracing and mass testing. Approximately 30 million PCR tests were performed between 26 May 2021 and 8 June 2021. The intense testing and screening of high-risk populations makes cryptic transmissions unlikely. Nearly all the infections we

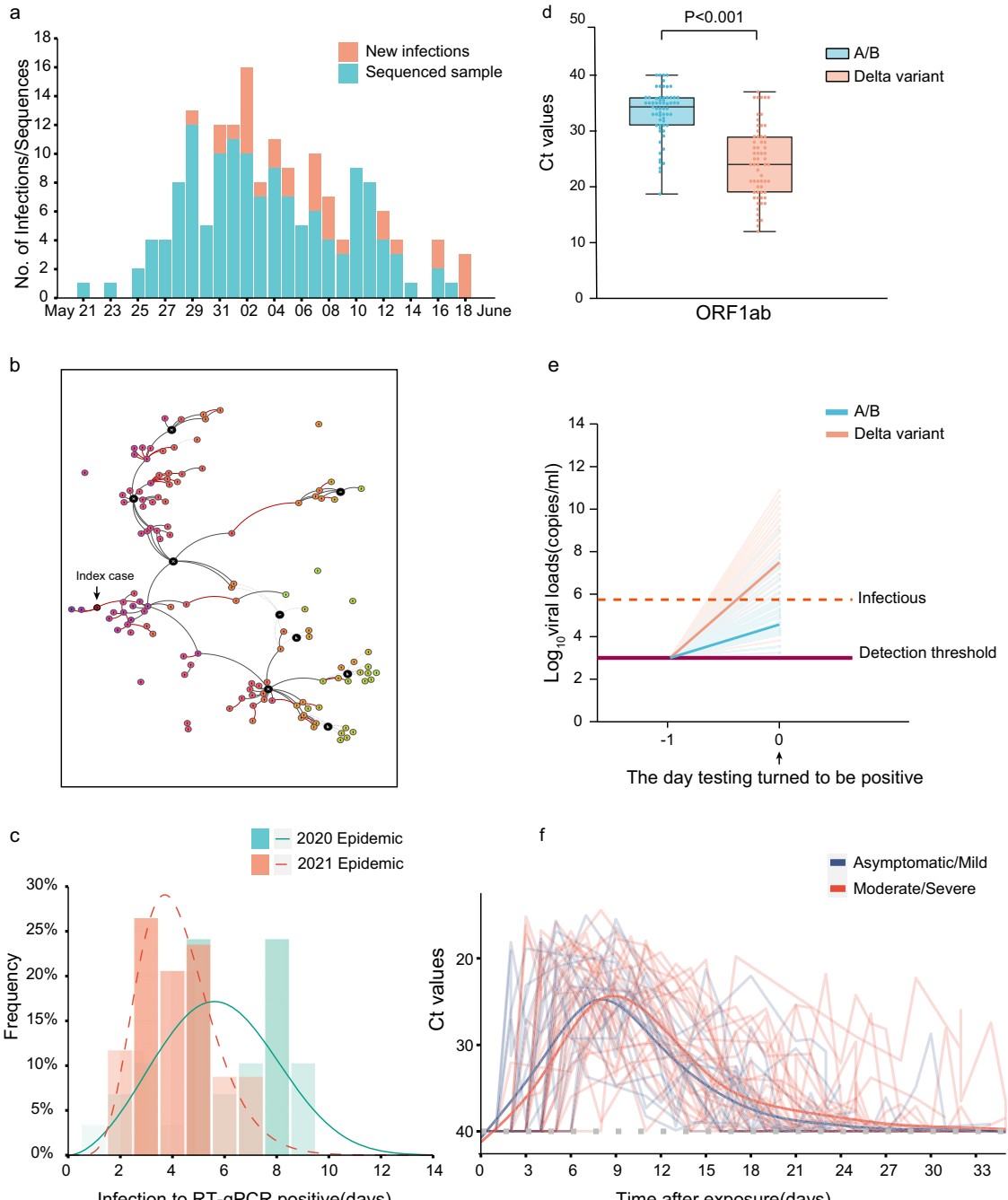

**Fig. 1 Summary of the epidemiology and early detection of the Delta SARS-CoV-2 variant in Guangdong. a** Time series of 167 laboratory-confirmed infections originating from the first index case on 21 May 2021. Daily numbers of new infections are shown in red and samples with high-quality sequences (coverage >95%) are shown in blue. **b** The Delta variant transmission in the Guangzhou outbreak. The transmission relationship between 126 sequenced cases was indicated with solid lines (high confidence) or the dash lines (unsure). The interactive version could be found at https://viz.vslashr.com/guangdongcdc/. **c** Estimate of the time interval between exposure and time of the first PCR + test in quarantined individuals. The curves show the best-fitting distributions of the interval durations for Delta variant cases ($n = 46$) and for A/B clade cases ($n = 29$). Bars show the histograms of estimated intervals durations (days). **d** Two-tailed Wilcoxon signed-rank test $P$ values ($P < 0.001$), showing significant changes in Ct (Cycle threshold) values of the first PCR + test in quarantined individuals, for the Delta variant infections ($n = 62$) and for previous A/B clade strains infections ($n = 63$). Dots represent Ct values for RT-PCR of the ORF1ab gene. Box plots indicate the median (middle line), 25th and 75th percentiles (box) and minimum and maximum value (whiskers). **e** Schematic of the relation between the viral growth rate and the relative viral loads on the day viruses were first detected (Day 0). The viral load on Day 0 was measured. The horizontal dashed line in purple represents the detection threshold of RT-PCR testing; the dashed line in red represents the lower limit above which infectious viruses could be potentially isolated. **f** Temporal profile of serial Ct value from quarantined individuals. Longitudinal PCR testing was performed on quarantined individuals ($N = 46$, overall and stratified by symptom status. The detection limit was Ct = 40. The thick lines show the trend in viral load, using smoothing splines. We only included data from the first PCR + test to the first PCR- test during the infection; the few re-positive tests were excluded (Supplementary data 1).

**Table 1 Latent interval and virus shedding interval of the Delta infections.**

| Interval | Asymptomatic/mild median (IQR) | Num. | Moderate/ severe median (IQR) | Num. | Overall median (IQR) | Num. | Sig. |
|---|---|---|---|---|---|---|---|
| Latent interval | 4.0 (2.5, 5.0) | 17 | 5.0 (4.0, 5.5) | 29 | 4.0 (3.0, 5.0) | 46 | $P = 0.097$[*] |
| Virus shedding interval | 16.0 (14.25, 19.5) | 12 | 18.0 (12.0, 22.0) | 19 | 17.0 (14.0, 20.0) | 31 | $P = 0.590$[**] |

Latent interval: from the exposure to first PCR+.
Shedding interval: from the PCR + to PCR−.
[*]Two-tailed Wilcoxon rank-sum test.
[**]Two-tailed two-sample t test.

identified could be connected epidemiologically, either through evidence of direct contact, or indirectly (staying in or visiting the same area) (Fig. 1b). In addition, all sequences could be genetically traced back to the index case. This provided a unique opportunity for us to characterize virus transmission dynamics at a finer scale, particularly the extent to which virus genetic diversity is transmitted among hosts. Whole-genome deep sequencing was performed on all identified infections, and 126 high-quality viral genomes (coverage>95%) were obtained, comprising 75% of identified infections in the outbreak (Fig. 1a).

Phylogenetic analysis was performed by combining the virus genomes we obtained from the Delta outbreak with genomes from 346 imported cases; the latter represents travelers to Guangdong during March 2020 to June 2021 who arrived from 66 different source countries. We also included a set of reference sequences, comprising 50 genomes randomly selected from each of 13 defined NextStrain clades (https://nextstrain.org/) and the notified VOCs (Alpha, Beta, Gamma, Delta). The viral lineage distribution of the imported cases was approximately representative of the SARS-CoV-2 genetic lineages that were circulating at that time at the global scale. These importations pose a challenge for disease control and prevention in Guangdong, China (Fig. 2a).

Viral phylogenies of the Guangzhou outbreak were inferred using the assembled consensus sequence of each sample, which was generated by choosing the majority-frequency nucleotide (>50%) at each position. All Guangzhou outbreak sequences were segregated into a single cluster (Fig. 2a). Compared with the index case (5137) of the outbreak, 34 substitutions were identified among 125 cases during the 26-days long outbreak (Fig. 2b). The most genetically divergent outbreak sequence contained four nucleotide differences from the index case sample. To understand how these variants emerged, grew, and finally fixed during the epidemic (and during the SARS-CoV-2 pandemic more generally), we estimated within-host virus diversity for each sample by mapping polymorphic sites against the consensus genome of the index case (XG5137_GZ_2021/5/21), thereby generating a list of intra-host single-nucleotide variants (iSNVs). Minor iSNVs were called by setting 3% as the threshold for minor allele frequency in order to exclude potential PCR/sequencing errors and minimize false discovery rates[12–14] (Supplementary Fig. 2). For 126 high-quality sequences, most samples harbored 2 iSNVs (median) which is consistent with other reported levels (Supplementary Fig. 3)[12,13].

We calculated the transmission bottleneck size among epidemiologically confirmed transmission pairs. Contact tracing and epidemiological investigation enabled us to assign 111 donor-recipient transmission pairs with a high degree of confidence. Of these, the donor had one or more iSNVs above the variant calling threshold of 3% in 60 transmission pairs (Supplementary Table 2), enabling estimation of the transmission bottleneck size, $N_b$, using the beta-binomial method[15]. The robustness of iSNVs calculation with the high-throughput sequencing method was closely related with the viral loads of samples (Supplementary Fig. 2). Good concordance was achieved in replicated sequencing for samples with relatively high viral loads (Ct ≤30), while a high proportion of

discordant iSNVs was observed in samples with relatively lower viral loads (Ct >30), possibly generated by PCR biases[16,17] (Supplementary Fig. 2c–h). Therefore, we only estimated bottleneck in the transmission pairs with samples Ct ≤30. Uncertainty in the $N_b$ estimate was large for some transmission pairs, with the 95% confidence interval ranging from 1 to ~500 or more, suggesting for some pairs the sequencing data was not sufficiently informative. Those transmission pairs with confidence intervals of estimated bottleneck over 1–100 were not shown. Finally, the maximum likelihood estimate for $N_b$ was one for 26 out of these 29 transmission pairs, and two or three for the remaining three transmission pairs (Fig. 3a). Our data suggest the transmission bottleneck of SARS-CoV-2 is very narrow in general, consistent with the previous household transmission studies[12,18]. The transmission bottleneck size influences the extent to which within-host diversity contributes to viral diversity at the population scale. The stringent transmission bottleneck of SARS-CoV-2 suggests the substitutions we observed in Guangdong outbreak (and SARS-CoV-2 pandemic more generally) largely resulted from de-novo mutations appearing within individuals.

Although the transmission bottleneck of SARS-CoV-2 is narrow in general, it may be not constant and could be impacted by both viral and host factors. To investigate the contribution of the transmission of minor iSNV to population-level diversity, we identified the sequences with minor iSNVs and the sequences in which the derived nucleotide state was fixed. Notably, sequences exhibited minor intra-host single-nucleotide variants (iSNVs) at 10 of the 34 variant sites (positions that varied from the sequence of the first index case) (Fig. 2b). The direct epidemiological links (transmission pair 1, 2, 3, 45) were observed between the hosts with the minor iSNVs and their recipients with these iSNVs fixed (Fig. 3b). Therefore, at least three fixed substitutions in this outbreak could be traced to the direct transmission of minor iSNVs. It is also noteworthy that the transmission pairs with 5137 as the donor had a relatively higher estimated $N_b$, suggesting heterogenicity in iSNV transmission (Fig. 3a). The differences in bottleneck size are possibly due to the different transmission route or exposure doses, as has been observed for influenza[19]. The case 5137 presented a high viral load (Ct value of 17.6, ~2 × 10^9 copies/mL in oropharyngeal swabs) 2 days after their direct contact with the cases 5645 and 5571. The high viral loads, direct contacts and relatively high frequency of the iSNVs (4% for T21673C and 47% for C27086T) may have enabled the successful transmission of iSNVs to the recipients (Fig. 3b). Taken together, our observations suggest that the transmission bottleneck of SARS-CoV-2 is stringent in general, with most donor iSNVs not found in the recipients. However, transmission of minor iSNVs, with their fixation in the recipient host, resulted in at least some of the substitutions that accumulated during the outbreak.

## Discussion

Our study has several limitations. First, the sample size is limited because the study relied on intensive epidemiological investigation

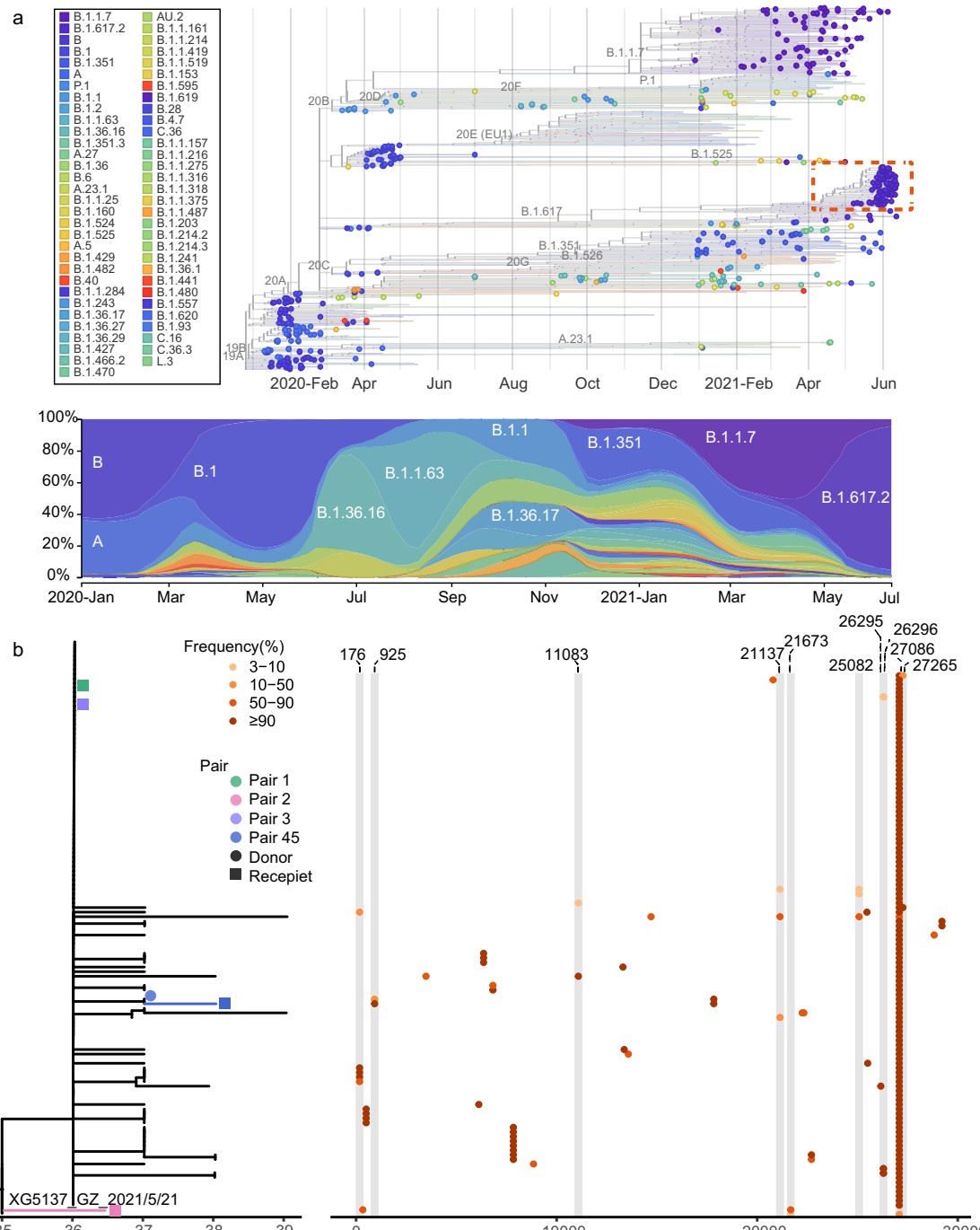

**Fig. 2 Viral phylogenies and transmission dynamics of the Guangzhou outbreak. a** A time-resolved phylogenetic tree was estimated using the NextStrain pipeline and includes (i) Guangdong sequences collected from local infections and imported cases, January 2020–June 2021, and (ii) reference sequences from different genetic lineages. The sequences from the Guangdong Delta variant outbreak (21 May 2021–18 June 2021) are highlighted with a red box. The changing frequencies of SARS-CoV-2 lineages (according to pangolin classification scheme, https://github.com/cov-lineages/pangolin) identified in Guangdong (most of which are imported) are shown in the lower panel. **b** Maximum likelihood tree of 126 sampled sequences of the Guangzhou outbreak. The sequence of the first index case (XG5137_GZ_2021/5/21) was used as the reference. SNV frequencies (%) across the virus genome (numbering according to the reference) are marked with colored dots (right hand panel).

to identify potential infections and their exposure times, strict quarantine to exclude other possible co-infections, and daily PCR testing to determine when virus shedding first become detectable and viral trajectories. Second, we only compare the relative viral loads of Delta infections to A/B lineage infections during initial epidemic. Since Guangdong had been free of any COVID-19 outbreaks for the previous 14 months[5], the clinical surveillance data from other SARS-CoV-2 lineages is limited. The A/B SARS-CoV-2 lineages (Supplementary Table 1), previously circulating in Guangdong (also represented as the early strains circulating in China) did not carry the D614G spike mutation, which has been associated with increased viral loads and transmissibility[20].

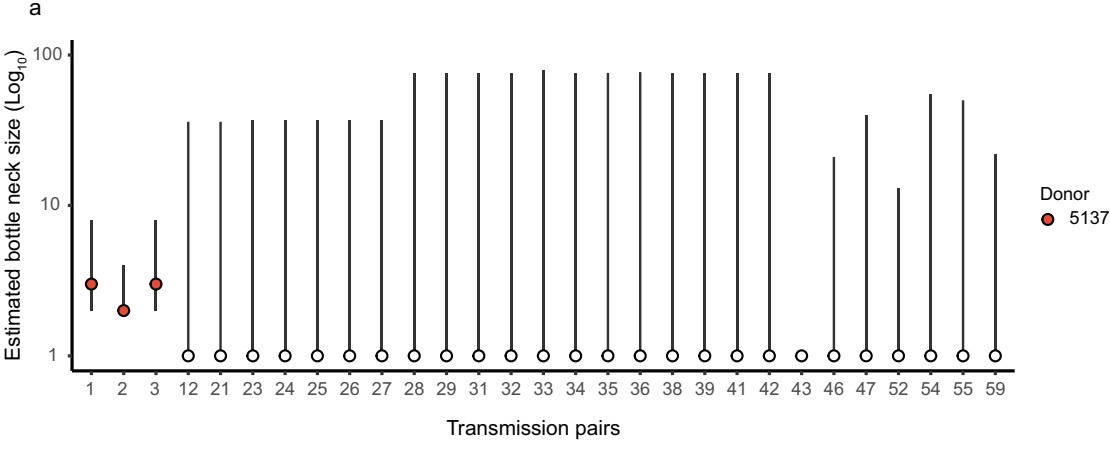

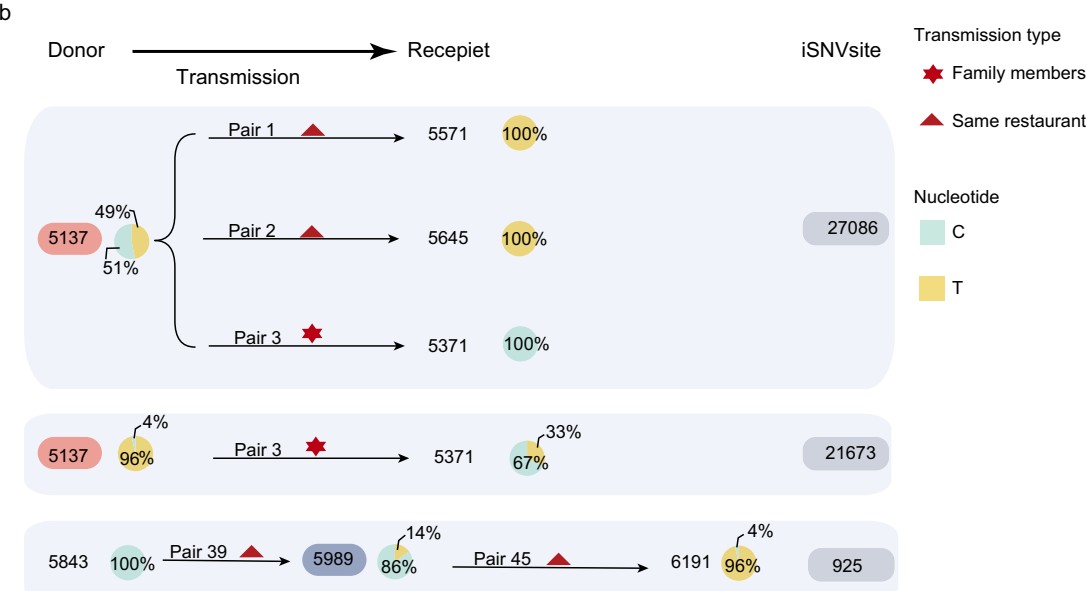

**Fig. 3 Transmission bottleneck size and transmissions of intra-host variants in the outbreak. a** Estimated bottleneck size in donor-recipient transmission pairs ($n = 29$) was calculated using the exact beta-binomial method. Bars show mean and 95%CI, calculated using the likelihood ratio test. **b** Minor iSNVs transmission resulted in the diversity of viral population. The pie charts show the frequency of iSNVs. Arrows show the direction of transmission for those pairs of cases for which this is known with high confidence.

Therefore, the 1000× higher viral loads at first positive sample when comparing Delta to A/B may be less if compared to 614G SARS-CoV-2 strains infections including the Alpha variant.

Notably, this cohort study on quarantined cases has a different study design compared to the cross-sectional investigations in most other studies, enabling estimation of the latent interval, viral shedding interval, and viral load on the day the virus was first detectable. The ~1000 times higher viral loads observed on the day of the first PCR + test for Delta infections indicate the high transmission risk of the Delta variant in early stage of infections. How are the peak viral loads of the Delta infections compared to other SARS-CoV-2 strains required more longitudinal studies. Our dense sampling on 46 individuals indicates the peak viral loads (corresponding to the lowest Ct value) in the Delta infections were around $10^9$ copies/mL in oropharyngeal swabs which provides the baseline information for the follow-up studies on the Delta or other emerging SARS-CoV-2 variants. Third, the relative viral loads, calculated from the Ct values of qualitative RT-PCR, might be affected by other factors such as sampling procedure or

batch effects. To control for batch effects, internal controls were set for each PCR experiment (see details in Methods section). Sampling procedures were similar between the 2020 and 2021 epidemics, and there is no reason to believe they changed systematically. Finally, during the outbreak, we used the multiplex PCR with high-throughput sequencing to provide timely genetic sequences for tracing the viral transmission. All clinical samples were detected and sequenced in real-time, meaning that sample sequencing could not be performed in replicate or triplicate during the outbreak. Grubaugh et al.[21] have shown that amplicon-based sequencing provides comparable accuracy to metagenomic sequencing in iSNV analysis and duplicate sequencing could remove a few false-positive iSNVs with frequency >3%[21]. In this study, we re-sequenced 15 patient samples and sequenced mixtures with known ratios of viral lineages. Our results also confirmed the robustness of the amplicon-based sequencing in calling iSNVs in relatively higher viral load samples (Ct ≤ 30), with concordance in iSNV frequencies above 3% found in most replicates (Supplementary Fig. 2f–h). However, without

technical replicates for all samples, we cannot exclude the possibility that a few false-positive iSNVs in donor samples may have led to bottleneck underestimation for some transmission pairs.

In this study, we characterized a large transmission chain that originated from the first local infection of the SARS-CoV-2 Delta variant in mainland China. We find evidence for a potentially higher viral replication rate of the Delta variant, as viral loads in Delta infections are ~1000 times higher than those for clade A/B infections on the day of the first PCR + test. This suggests that infectiousness of Delta variant during the early stage of infection is likely to be higher. Consequently, the frequency of population screening should be optimized[22]. If Delta-infected patients are indeed more infectious during the pre-symptomatic phase, then timely quarantine (before clinical onset or PCR screening) for suspected cases or for close contacts becomes more important. At the time of writing, the SARS-CoV-2 Delta variant is now widely spread, even in regions with a high vaccination rate. Compared to Beta, the Delta variant is less able to escape the prior immunity[23,24]. The relatively high viral titres detected in the pre-symptomatic phase of infection, even on the day of the first PCR + test, may explain its extraordinary transmissibility compared to other variants. Frequent breakthrough infections also raise the concern that Delta variant viruses could adapt within individuals to evade acquired immunity, with the subsequent transmission of immune-escape iSNVs as the virus spreads. Our study suggests that although the transmission bottleneck of SARS-CoV-2 is narrow in general, the transmission of minor iSNVs explains some of the fixed substitutions observed in the virus population during the outbreak. In some settings, advantageous iSNVs could arise and become fixed within one transmission generation, with further spread in the virus population if the epidemic is not well contained. SARS-CoV-2 virus is continuously evolving, potentially adapting, and globally transmitted. New SARS-CoV-2 variants will emerge through the time. The viral profile of the Delta variant we provided here is valuable for the comparative study on the infectivity of other emerging variants and then evaluating their epidemic risks in the future.

## Methods

**Ethics**. This study was approved by the institutional ethics committee of the Guangdong Provincial Center for Disease Control and Prevention (GDCDC). Written consent was obtained from patients or their guardian(s) when samples were collected. Patients were informed about the surveillance before providing written consent, and data directly related to disease control were collected and anonymized for analysis.

**Identification and quarantine of contacts**. We followed the same quarantine scheme in both 2020 and 2021 SARS-CoV-2 outbreaks. Once a case was detected as SARS-CoV-2 RNA positive, the individual was reported as an index case and isolated, and the Guangdong CDC and local CDCs conducted an immediate field investigation. The exposure history for positive cases and their close contacts were obtained through an interview, public video monitoring systems and cell phone apps, etc. The index cases' trajectories were made publicly available in around 24 h after identification. Information on the demographic distribution of SARS-CoV-2 cases can be found at the website of the Health Commission of Guangdong Province (http://wsjkw.gd.gov.cn/xxgzbdfk/yqtb/ and https://github.com/Jinglu1982/Delta-variant-outbreak-in-GZ). The close contacts of the index cases were traced and centrally isolated. In the 2020 epidemic, the PCR testing was generally performed on the 1, 4, 7, and 14 days after the isolation according to the Diagnosis and Treatment Scheme for Covid-19 released by the National Health Commission of China (Version 7). Only the 29 quarantined cases who had daily PCR testing were included in this study (Fig. 1c). In the 2021 epidemic, the Delta infections presented a shorter latent interval (from exposure to virus shedding becoming detectable) and we performed daily PCR testing during the 14 days after exposure for all quarantined cases until the infection was confirmed. The confirmed cases were transported to sentinel hospitals and longitudinal testing was performed on 46 individuals (Fig. 1f).

**PCR testing and quantification**. Since the first local SARS-CoV-2 infection was reported on May 21 in the capital city of Guangdong, the enhanced surveillance was performed by Guangdong CDC and local CDCs to detect suspected infections. Epidemiological investigations had been done on all confirmed cases. Population

screening was performed by third-party detection institutions. Once virus-positive samples were confirmed, the oropharyngeal swab samples preserved in 3 mL of viral transport medium were sent to Guangdong CDC in 24 h. To make the results comparable, in Guangdong CDC, the real-time reverse transcription PCR (RT-PCR) were performed by using the same commercial kit (DA0931, DaAn Gene) and RT-PCR machine (CFX96, BioRad) as the previous studies[5,25]. For quality control, each time the clinical samples were tested, the aliquoted positive and negative controls of the kit were performed RNA extraction and RT-PCR along the samples. The positive control was constructed pseudovirus including the RT-PCR targeting fragments. We assumed the copies of the aliquoted positive control were the same for each RT-PCR experiment. To ensure the results from each RT-PCR experiment were comparable, the threshold line was adjusted to guarantee the Ct of the positive control was the same for each experiment (setting as 32 for the kit in this study). For viral load quantification, the reference pseudovirus (Reference Material no. GBW(E)091132, Guangzhou BDS Biological Technology) including the fragment sequences of SARS-CoV-2 (MN908947.3, https://www.ncbi.nlm.nih.gov/nuccore/MN908947) was used, of which the absolute copies of ORF1ab were determined by digital RT-PCR. The pseudovirus was ten-fold serially diluted (from $1.67 \times 10^7$ to $1.67 \times 10^3$ copies/mL), and viral RNA was extracted and PCR quantified to build a standard curve (Supplementary Fig. 4). The relative viral loads in oropharyngeal swab samples were converted to RNA copies per mL according to the Ct value of the ORF1ab gene.

The surge population screening test ensure all possible infections were identified and 111 donor-recipient transmission pairs were assigned with very high confidence. All transmission pairs met the following rules: 1. The recipient was the close contract of the donor and had a clear and direct epidemiological link to the donor; 2. The recipient did not have any contacts with other identified cases. The unsure epidemiological link (marked with dash line in Fig. 1b) means we did not find directly contacts between cases but they are potentially associated according to their trajectories like living in the same building or visiting the same market.

**Modelling**. We present SARS-CoV-2 viral loads in the oropharyngeal swabs of quarantined individuals by days between exposure and the time of RT-PCR test. A smoothing spline was fitted to the Ct values to summarize the overall trend, adding visualization. Specifically, a generalized additive model (GAMs), $E(Y) = \beta 0 + s(t)$, with an identity link was fitted, where Y are the Ct values, β0 is the intercept and s(t) is a cubic spline evaluated at t days after exposure. We also compared the viral loads by the symptom status defined according to the Diagnosis and Treatment Scheme for Covid-19 released by the National Health Commission of China (V8).

**Viral isolation**. Vero E6 cells were used for SARS-CoV-2 virus isolation and passage. The cells were inoculated with 100 μl processed oropharyngeal swab samples. Cytopathic effect (CPE) was observed daily. If there was no CPE observed, cell lysis was collected by centrifugation after three repeated freeze-thaw cycles and 100 μl supernatant was used for the second round of passage.

**Virus amplification and sequencing**. Total RNAs were extracted from oropharyngeal swab samples by using QIAamp Viral RNA Mini Kit (Qiagen, Cat. No. 52904). Virus genomes were generated by two different approaches, (i) using commercial sequencing kit of BGI (ATOPlex 1000021625) and sequencing on the BGI MGISEQ-2000 ($n = 12$), and (ii) using version 3 of the ARTIC COVID-19 multiplex PCR primers (https://artic.network/ncov-2019) for genome amplification. cDNA (2.5 μL) was amplified in two multiplexed PCR reactions (26 rounds) followed by library construction with Illumina Nextera XT DNA Library Preparation Kit and sequencing with PE150 ($n = 76$) or SE100 ($n = 38$) on Illumina Miniseq. We report only high-quality genome sequences for which we were able to generate >95% genome coverage.

**Sequence analysis**. The bioinformatics pipeline for BGI platform (https://github.com/MGI-tech-bioinformatics/SARS-CoV-2_Multi-PCR_v1.0) was used to generate consensus sequences and call single-nucleotide variants relative to the reference sequence. For sequence data from Miniseq, the raw data were first quality controlled (QC) using fastp 0.20.1[26] to trim artificial sequences (adapters), to cut low-quality bases (quality scores < 20). The qualified reads were mapped to the first index case (XG5137_GZ_2021/5/21) using BWA 0.7.17[27]. The bam files were sorted by SAMtools 1.7[28] and the PCR primers were trimmed by using iVar 1.3.1[21]. The consensus sequences were determined with iVar 1.3.1[21], taking the most common base as the consensus (allele frequency >50%). An N was placed at positions along the reference with the sequencing depth fewer ≤10. The surge population screening test ensure all possible infections were identified and through the contact tracing the donor-recipient transmission pairs could be assigned with high confidence. To characterize the viral transmission in these pairs, we identified iSNVs relative to the reference genome (XG5137_GZ_2021/5/21) for each sequence with iVar 1.3.1 using the following parameters: alternated frequency at a SNV site ≥3%; total sequencing depth at SNV site ≥100; sequencing depth for the variant allele ≥10; iVar PASS = TRUE. We exclude the head and tail sequences of viral genome (corresponding to the positions 1 to 100 and 29803 to 29903 in Wuhan-Hu-1 reference genome) due to the lower sequencing coverage for most samples in the analysis and the 6 "highly shared" iSNV sites (1959, 4091, 21987, 24404, 28389, 28448) were excluded (those observed in 10 or more samples across the entire dataset)[12]. To infer the iSNVs transmission in donor-recipient pairs, all sites with ≥3% minor allele frequency in the assumed donor were used in the

analysis. In the recipient, all reads at these sites were considered, with a variant calling threshold of 3% using the beta-binomial method of Sobel Leonard et al.[15]. The next-strain pipeline[29] was used to analyze and visualize the genetic distribution of SARS-CoV-2 infections and its dynamic change in Guangdong between January 2020 and June 2021. Maximum likelihood (ML) tree was estimated with phyml 3.0[30] using the HKY + Q4 substitution model with gamma-distributed rate variation[31]. The branch length was recalculated as the number of mutations to the reference sequence of the first index case. The tree was visualized with R package of ggtree 2.4.1[32].

**Reporting summary**. Further information on research design is available in the Nature Research Reporting Summary linked to this article.

## Data availability

All sequencing reads mapped to the reference sequence (the sequences of the first index case, XG5137_GZ_2021/5/21) have been deposited to the GSA database of National Genomics Data Center (https://bigd.big.ac.cn/) with submission number CRA004896 (https://ngdc.cncb.ac.cn/gsa/browse/CRA004896). The generated consensus sequences were submitted with accession number GWHBDIM01000000 – GWHBDNH01000000 (https://ngdc.cncb.ac.cn/gwh/jbrowse/index), and also shared to NCBI GeneBank with accession number OL663920 – OL664045 (https://www.ncbi.nlm.nih.gov/nuccore). The data underlying Supplementary Figure 4 is provided in Supplementary data 1.

## Code availability

The pipeline for sequencing data analysis was deposit in https://github.com/Jinglu1982/Delta-variant-outbreak-in-GZ. Code to implement the beta-binomial method is publicly available[15].

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

## Acknowledgements

We gratefully acknowledge the efforts of China national CDCs, Guangdong local CDCs, hospitals, and the third-party detection institutions in epidemiological investigations, sample collection, and detection. J.L. was supported by Science and Technology Planning Project of Guangdong (2018B020207006), the Key Research and Development Program of Guangdong Province (2019B111103001); B.L., J.W., and J.L. were supported by Guangdong Workstation for Emerging infectious Disease Control and Prevention, Chinese Academy of Medical Sciences (2020-PT330-004); J.H. and J.L. were supported by China Evergrande Group (2020GIRHHMS11). We thank Kaiyuan Sun from National Institutes of Health for insightful comments. The views expressed in this article are those of the authors and not necessarily those of the Guangdong Provincial Center for Diseases Control and Prevention, or the Guangdong Provincial Institute of Public Health.

## Author contributions

J.L., J.H., Y.L. and J. Yuan. designed the study. B.L., A.D., K.L., Y.Hu., Z.L., Y.S., Q.G., L.Z., H.Zhang., M.Z., F.O., J.Su., J.Xu., H.L., P.Z., T.H., M.L., Y.Z., H. Zheng., H. Zeng., Z. Zheng., Y.Huang., J. Yu. and L.Y. undertook fieldwork and experiments. W.S., R.C., M.K., M.T., J.W., J.C., H. Zhong., X.D., M.K. and J.H. provided epidemiological information. J.L., Q.X., Z.L., J.P., J.Xiao., T.L., M.H. and K.L. designed and performed genetic analyses. Q.X., J.Sun., H.J., J.Xiao. and T.L., performed epidemiological analyses. J.L., H.J., Q.X. and J.Sun. drafted the manuscript. K.L. and O.G.P. edited the manuscript. All authors contributed to the manuscript. All authors read and approved the contents of the manuscript.

## Competing interests

The authors declare no competing interests.
