## [Peer Review File · Nature Communications]

Viral infection and transmission in a large, well-traced outbreak caused by the SARS-CoV-2 Delta variantEditorial Note: This manuscript has been previously reviewed at another journal that is not operating a transparent peer review scheme. This document only contains reviewer comments and rebuttal letters for versions considered at *Nature Communications*.

REVIEWER COMMENTS

Reviewer #1 (Remarks to the Author):

Summary

Li and colleagues provide an in-depth description of a single chain of transmission of the SARS-CoV-2 delta variant in mainland China. They combine detailed epidemiological tracking and diagnostic data to deep sequencing of virus genomes to fully characterize the Delta variant transmission throughout the outbreak and compare it to earlier A/B lineages transmission chains. They find evidence suggesting that the Delta variant is more infectious and replicates faster during the early stages of infection compared to early pandemic lineages, with a small bottleneck size upon transmission (1 to 3 virions).

Recommendation

The authors have provided additional quality assessments of their iSNVs identification procedure but I still have concerns regarding the quality of iSNV data that would need to be addressed prior to publication (see major comments).

Major comments

iSNVs identification

I still have concerns regarding the iSNV data generated and analyzed in this study. In the previous rebuttal letter, the authors shared resequencing data for 15 of the samples, showing a good proportion of iSNVs above 3% and below 10% were not found in both replicates, suggesting an important bias. I suggested removing samples with Ct values above 30 from the analysis set but in their response to the reviews, the authors do not mention whether removing these samples allowed to remove artefactual iSNVs that were not found in sequencing replicates. Could the authors please share this information? I certainly appreciate the fact that the authors added a sequencing and iSNVs identification quality assessment (Figure S2), but since the bottleneck size estimates rely on few samples (for most transmission pairs uncertainty is above an order of magnitude) and thus, few iSNVs, it is critical to confirm the robustness of this data.

In the response to the reviewers, the authors mention that they “also checked that the iSNVs in transmission bottleneck analysis were not resulted from the reads having primer mismatches.” After checking the methods section and the pipeline code, I do not find any trace of this step having been added to the pipeline. Could the authors please point me to the relevant section/supplementary materials?

Bottleneck size estimates

“Uncertainty in the N_b estimate was large for some transmission pairs, with the 95% confidence interval ranging from 1 to ~500 or more, suggesting for some pairs the sequencing data was not sufficiently informative.”

Given the data shown in Figure 2c, uncertainty is large for most transmission pairs. I suggest focusing on the 15 transmission pairs for which uncertainty is within a single order of magnitude and include only these in the bottleneck size estimation.

Minor comments

Introduction section

It would be good to clarify what the status of 'unsure' cases really is. Have these cases not been linked to other cases identified in the outbreak, or do the epidemiological and genetic links disagree?

"We discuss how intervention strategies may need to be adjusted to cope with the virological properties of this emerging variant."

Results section

I am not sure what the authors mean by "virological properties", but I guess they refer to the changes in latent interval, viral shedding interval and viral load trajectories they report in the manuscript. I would suggest being more precise about the properties in question.

Methods section

Is there an english version of the web page showing the demographics of SARS-CoV-2 cases, I could not find it using the link provided in the manuscript (<http://wsjkw.gd.gov.cn/xxgzbd/fk/yqt/b/>).

Discussion section

- "If Delta infections are indeed more infectious during the pre-symptomatic phase,"

Please rephrase, I assume the authors meant "If Delta-infected patients are indeed more infectious...?"

- "Compared to Beta and Gamma, the Delta variant is less able to escape the prior immunity."

I think a reference is needed here.

Dear Editor,

We really appreciate the reviewer's nice comments and guidance on iSNVs analyzing procedure which increase the robustness of the study and provide us some important notes in analyzing the iSNVs sequencing data. We have addressed the reviewer's all requests in the revised paper. The revisions were highlight in red in the main text. Below we provide a point-by-point response.

Jing Lu

REVIEWER COMMENTS

Reviewer #1 (Remarks to the Author):

Summary

Li and colleagues provide an in-depth description of a single chain of transmission of the SARS-CoV-2 delta variant in mainland China. They combine detailed epidemiological tracking and diagnostic data to deep sequencing of virus genomes to fully characterize the Delta variant transmission throughout the outbreak and compare it to earlier A/B lineages transmission chains. They find evidence suggesting that the Delta variant is more infectious and replicates faster during the early stages of infection compared to early pandemic lineages, with a small bottleneck size upon transmission (1 to 3 virions).

Recommendation

The authors have provided additional quality assessments of their iSNVs identification procedure but I still have concerns regarding the quality of iSNV data that would need to be addressed prior to publication (see major comments).

Major comments

iSNVs identification

I still have concerns regarding the iSNV data generated and analyzed in this study. In the previous rebuttal letter, the authors shared resequencing data for 15 of the samples, showing a good proportion of iSNVs above 3% and below 10% were not found in both replicates, suggesting an important bias. I suggested removing samples with Ct values above 30 from the analysis set but in their response to the reviews, the authors do not mention whether removing these samples allowed to remove artefactual iSNVs that were not found in sequencing replicates. Could the authors please share this information? I

certainly appreciate the fact that the authors added a sequencing and iSNVs identification quality assessment (Figure S2), but since the bottleneck size estimates rely on few samples (for most transmission pairs uncertainty is above an order of magnitude) and thus, few iSNVs, it is critical to confirm the robustness of this data.

Response: Thanks for your good suggestions. The new Figure S2 showed the concordance of replicated sequencing on samples with lower viral loads ($CT > 30$) and samples with relative higher viral loads ($CT \leq 30$). Consistent with the result in Figure S2b and other's study, a high concordance was achieved in samples with relative higher viral loads, indicating some PCR bias may generated for samples with lower viral copies (Figure S2c-e). Thanks again for this valuable suggestion. we revised the Figure 3c by investigating on 29 transmission pairs in which the samples had CT value ≤ 30 and the confidence interval of estimated bottleneck was not over 1-100. We also revised the description in main text as “The robustness of iSNVs calculation with this high-throughput sequencing method was closely related with the viral loads of samples (Figure S2). For samples with $Ct \leq 30$, a relative high concordance could be achieved from the replicated sequencing and PCR biases were more likely to be generated in sample with relative lower viral loads (Figure S2 c-h). Therefore, we only estimated bottleneck in the transmission pairs with samples $Ct \leq 30$. Uncertainty in the N_b estimate was large for some transmission pairs, with the 95% confidence interval ranging from 1 to ~500 or more, suggesting for some pairs the sequencing data was not sufficiently informative. Those transmission pairs with confidence interval of estimated bottleneck over 1-100 were not shown. Finally, the maximum likelihood estimate for N_b was one for 26 out of these 29 transmission pairs, and two or three for the remaining 3 transmission pairs (Figure 2c).” Line 184-195.

Figure S2(c-h) Fifteen replicate pairs were performed separate reverse transcription (RT), PCR amplification, and library preparation steps. Results were shown with sequencing pairs Ct >30 (6 pairs, c-e) and Ct ≤30 (9 pairs, f-h), respectively. We included sites with alter frequency ≥3%, depth >100 and alter depth >10 in at least one of the 30 replicates. The points represent the iSNVs identified in all replicate pairs. If minor iSNVs are reproducible, we expect a positive correlation. The blue line shows 3% iSNVs frequency.

In the response to the reviewers, the authors mention that they “also checked that the iSNVs in transmission bottleneck analysis were not resulted from the reads having primer mismatches. “ After checking the methods section and the pipeline code, I do not find any trace of this step having been added to the pipeline. Could the authors please point me to the relevant section/supplementary materials?”

Response: Sorry we don't describe this clearly. Since all the infection cases in this outbreak could be traced back to the first index case. We first analyzed the primer mismatches by aligning the primers to the genome of the first index case. Three mismatches (21987, 24404, 28448) were identified between artic V3 primers and the genome sequences of the first index case (see attached figure below). In the sequence analysis, we followed the previous reviewer2's suggestion by using iVar to trim potential primer sequences. In addition, the potential iSNVs caused by these three mismatch sites were also excluded from transmission bottleneck analysis (Line 391-392).

Figure. The artic V3 primers were mapped to the genome sequence of the first index case (XG5137_GZ_2021/5/21). Three mismatches were identified in 21987, 24404, and 28448. All these three sites were excluded from the final iSNVs transmission analyses.

Bottleneck size estimates

“Uncertainty in the Nb estimate was large for some transmission pairs, with the 95% confidence interval ranging from 1 to ~500 or more, suggesting for some pairs the sequencing data was not sufficiently informative.”

Given the data shown in Figure 2c, uncertainty is large for most transmission pairs. I suggest focusing on the 15 transmission pairs for which uncertainty is within a single order of magnitude and include only these in the bottleneck size estimation.

See response above. In revised Figure 2c, we focused on the 29 transmission pairs with a relative higher viral loads and confidence interval of estimated transmission bottleneck within 1-100.

Minor comments

Introduction section

It would be good to clarify what the status of 'unsure' cases really is. Have these cases not been linked to other cases identified in the outbreak, or do the epidemiological and genetic links disagree?

We included the description in revised paper as “The unsure epidemiological link (marked with dash line in Figure 1b) means we did not find directly contacts between cases but they are potentially associated according to their trajectories like living in the same building or visiting the same market.” Line 340-343.

“We discuss how intervention strategies may need to be adjusted to cope with the virological properties of this emerging variant.”

I am not sure what the authors mean by “virological properties”, but I guess they refer to the changes in latent interval, viral shedding interval and viral load trajectories they report in the manuscript. I would suggest being more precise about the properties in question.

Corrected as “We discuss how intervention strategies may need to be adjusted to cope with the changes in viral shedding interval and viral load trajectories of this emerging variant.”

Methods section

Is there an english version of the web page showing the demographics of SARS-CoV-2 cases, I could not find it using the link provided in the manuscript (<http://wsjkw.gd.gov.cn/xxqzbd/fk/yqtb/>).

Sorry, there is no English version of this web page. We attached original data corresponding to Figure 1A in <https://github.com/Jinglu1982/Delta-variant-outbreak-in-GZ> , and the demographic information for each infection case could be found at <https://viz.vslashr.com/guangdongcdc/>.

Discussion section

“If Delta infections are indeed more infectious during the pre-symptomatic phase,”

Please rephrase, I assume the authors meant “If Delta-infected patients are indeed more infectious...”?

Corrected.

“Compared to Beta and Gamma, the Delta variant is less able to escape the prior immunity.”

I think a reference is needed here.

Thanks. We have included the relative reference in the revised paper (reference 22 and 23).

REVIEWERS' COMMENTS

Reviewer #1 (Remarks to the Author):

The manuscript has improved and most of my comments have been addressed. However, I remain concerned about the identification of iSNVs using amplicon sequencing without having triplicate sequencing for all samples. Based on the iSNV data from resequenced samples shared by the authors, excluding high Ct samples removes some – but not all – discordant iSNVs with a frequency above 3%. At this stage, the only way of ensuring the iSNV data is robust would simply be by confirming all iSNVs using resequencing in triplicate as described for example in Grubaugh et al. (<https://pubmed.ncbi.nlm.nih.gov/30621750/>).

Dear editor,

Thank you for reviewing our paper. We are pleased that for the most part the reviewers are happy with our paper, with the only outstanding point the number of replicate sequences. The following is our response to this point.

Reviewer #1 (Remarks to the Author)

The manuscript has improved and most of my comments have been addressed. However, I remain concerned about the identification of iSNVs using amplicon sequencing without having triplicate sequencing for all samples. Based on the iSNV data from resequenced samples shared by the authors, excluding high Ct samples removes some – but not all – discordant iSNVs with a frequency above 3%. At this stage, the only way of ensuring the iSNV data is robust would simply be by confirming all iSNVs using resequencing in triplicate as described for example in Grubaugh et al. (<https://pubmed.ncbi.nlm.nih.gov/30621750/>).

Response: To confirm the robustness and accuracy of the iSNV detection in this study, we have already **resequenced** patient samples **and sequenced** mixtures with known ratios of viral lineages (Figure S2). These results confirmed the robustness of the sequencing method in calling iSNVs in relatively higher viral load samples ($CT \leq 30$), with concordance in iSNV frequencies above 3% found in most samples (Figure S2 f-h). Comparing the duplicate sequencing of patient samples, we identified 27 iSNV sites with frequency above 3% in at least one replicate. Although seven of these were discordant (found above 3% in one of two replicates), five of these discordant sites were identified in the re-sequencing of a single sample (sample 6002). This sample has a relatively higher viral load (CT 28) and we suspect the freeze-thawing of the sample that occurred between the first and second rounds of sequencing is the cause for the detection of the iSNVs when re-sequenced. Excluding this sample, only 2 of 27 sites were discordant.

As is common for SARS-CoV-2, most of clinical samples are now depleted because they were used for PCR diagnosis as well as for the sequencing. As we described in our early response letter, all clinical samples from this outbreak were detected and sequenced in real-time to trace the outbreak and confirm the transmission chains, meaning we could not perform the sequencing in replicate and triplicate during the outbreak. To address reviewer's queries, we re-sequenced samples where possible to demonstrate the robustness of the method, and we think it unreasonable to only now request triplicate sequencing. Even if this were possible, it would require another round of freeze-thaw. Moreover, triplicate sequencing is extremely unusual, and as far as we know has not been done for SARS-CoV-2. Examples of other high-profile SARS-CoV-2 papers that used single or duplicate sequencing (but not triplicate) for amplicon data are: Popa et al Sci Trans Med 2021 (10.1126/scitranslmed.abe2555), Braun et al PLoS Path 2021 (<https://doi.org/10.1371/journal.ppat.1009849>), Tonkin-Hill et al eLife 2021 (<https://doi.org/10.7554/eLife.66857>), Valesano et al 2021 (10.1371/journal.ppat.1009499). Most of these studies used duplicate sequencing to check the robustness of their methods, rather than for all samples, for the reasons we have mentioned.

We thank the reviewer for suggesting the Grubaugh et al paper, which did use triplicate

sequencing to identify the best protocols for iSNV analysis. They didn't find any benefit in triplicate sequencing compared to duplicate sequencing for Illumina data (see their figure 3C), and in the protocol published with their paper they recommend duplicate sequencing: https://static-content.springer.com/esm/art%3A10.1186%2Fs13059-018-1618-7/MediaObjects/13059_2018_1618_MOESM3_ESM.pdf

Given these points, we hope you will agree that triplicate is unnecessary. We also discussed the reason why we did not perform duplicate sequencing for all clinical samples during the outbreak and the potential affects to the result in the revised manuscript (marked in red).

Best wishes,

Jing Lu